# Bridging the Gap: Dental Students’ Attitudes toward Persons with Disabilities (PWDs)

**DOI:** 10.3390/healthcare12141386

**Published:** 2024-07-11

**Authors:** Faris Yahya I. Asiri, Marc Tennant, Estie Kruger

**Affiliations:** 1Department of Preventive Dental Sciences, College of Dentistry, King Faisal University, Al-Ahsa 31982, Saudi Arabia; 2International Research Collaboration—Oral Health and Equity, School of Allied Health, The University of Western Australia, Perth, WA 6009, Australia; marc.tennant@uwa.edu.au (M.T.); estie.kruger@uwa.edu.au (E.K.)

**Keywords:** dental education, attitude of health personnel, dental students, persons with disabilities, dental health service, access to healthcare, Saudi Arabia

## Abstract

Background: Persons with disabilities (PWDs) face significant barriers in accessing dental care, resulting in poorer oral health outcomes compared to the general population. To reduce dental healthcare disparities, dental professionals must develop positive attitudes and acquire the necessary skills to treat PWDs effectively. This study aimed to assess the experiences, education, training levels, and attitudes of dental students and interns toward PWDs, and to explore the relationships between their exposure to PWDs, their education/training regarding managing PWDs, and their attitudes toward PWDs. Methods: Participants were recruited using convenience sampling. From May to June 2023, a pretested, validated online questionnaire (developed based on existing survey instruments and modified to fit the study context) was distributed to 98 eligible dental students and interns at King Faisal University. Descriptive and analytical statistics were used for analysis. Results: The response rate was 88.78%. There was a statistically positive correlation between students’ exposure to PWDs and their attitudes (*p* < 0.05). Statistically significant relationships were found between students’ education/training and their attitudes towards educational experiences and instructors (*p* < 0.01) and interpersonal interactions with PWDs (*p* < 0.05). No statistically significant relationship existed between the year of study and future encounters with PWDs (*p* = 0.176). Additionally, 42.5% of students felt inadequately prepared to treat PWDs, and 88.5% expressed a desire for further education. Notably, 20.7% reported no training on PWDs, while 70.1% preferred clinical education. Conclusions: The study underscores the importance of exposure and training in shaping dental students’ attitudes toward PWDs. Dental schools should prioritize structured teaching, greater contact with the disabled community, and hands-on experiences to ensure dental professionals are both well prepared and positively inclined to treat PWDs. These findings have significant implications for improving dental education curricula.

## 1. Introduction

According to the World Health Organization (WHO), disabilities result from the interaction of health conditions with personal and environmental factors, with an estimated 16% of the global population affected [1]. Within this context, the latest data from Saudi Arabia indicates a disability prevalence rate of 7.1%, and this rate is projected to rise further due to the ongoing increase in health risk factors, including obesity, physical inactivity, road accidents, and chronic diseases [2]. The United Nations defines persons with disabilities as those who have “long-term physical, mental, intellectual, or sensory impairments which, in interaction with various barriers, may hinder their full and effective participation in society on an equal basis with others” [3]. The Saudi law regulating the rights of persons with disabilities aligns with this definition and emphasizes the importance of accessibility, inclusion, and equal rights for all individuals with disabilities [4]. Persons with disabilities (PWDs) experience a disproportionate level of oral disease compared to the general population, which underlines the need for specialized and collaborative efforts to lower this risk [5,6]. Moreover, they face significant dental treatment barriers, including physical difficulty accessing healthcare facilities, financial barriers, insufficiently trained dental professionals, and communication or language barrier difficulties [6].

The greater unmet dental needs among PWDs compared to the general population may be attributed to the obstacles they face within the dental health system rather than being solely due to the nature of their disabilities [5,6,7]. Most dental care for PWDs is not overly complicated and can be effectively delivered in primary care and community environments, provided the dental professionals possess the appropriate attitude and skills [8]. However, biased perspectives among physicians toward PWDs contribute to healthcare disparities, impacting access to quality care and exacerbating existing inequalities [9].

Addressing attitudinal barriers, providing thorough training, and fostering an inclusive approach to healthcare can increase the likelihood of a general dental practitioner offering treatment in a community setting, which could help alleviate the strain on health service resources and minimize the need for referrals [10].

The World Dental Federation promotes the integration of special care dentistry training throughout all stages of dental education, including undergraduate programs [8].

Similarly, the International Association for Disability and Oral Health (IADH) has proposed a curriculum that would enable dentists to offer care to diverse populations upon graduation, and the 30 countries that reached this consensus have a high level of agreement regarding the treatment of PWDs [11]. Additionally, The Commission on Dental Accreditation (CODA) generates and upholds accreditation standards designed to continuously improve the quality of dental education programs, providing a service to both the public and dental professionals. As the national accreditor for these programs, CODA plays an essential role. A modification was made to the Accreditation Standard for Dental Education Programs in August 2019 and implemented in July 2020. One of the mandates, known as Standard 2-25, requires graduates from pre-doctoral dental training programs to possess the skills to evaluate and manage the treatment of PWDs [12].

In Saudi Arabia, where 74% of dentists are general practitioners [13], understanding undergraduate dental students’ attitudes toward and preparedness for treating PWDs is crucial. These practitioners often represent the first line to provide care for PWDs, highlighting the need to enhance dental training programs to prepare graduates to meet increasing demand.

In alignment with global efforts, Saudi Arabia’s Vision 2030 includes the Health Sector Transformation Program, which aims to upgrade healthcare by emphasizing public health and disease prevention and enhancing service quality and accessibility. This study seeks to understand how to improve the attitudes and preparedness of future dental health professionals, particularly dental students, concerning providing dental care for PWDs [14].

## 2. Methods

The study was approved by the Human Research Ethics Committee at the University of Western Australia (file reference—2022/ET000328 on 7 September 2022) and the Research Ethics Committee at King Faisal University, Al-Ahsa, Saudi Arabia. (approval number KFU—REC-2022-APR-EA000553 on 5 April 2022).

We designed a cross-sectional study to assess dental students’ experiences, education, and attitudes toward PWDs. The study was conducted at the College of Dentistry, King Faisal University between May and June 2023.

### 2.1. Study Participants

Participants were selected using convenience sampling based on their accessibility and willingness to participate. The initial pool included a total of 166 dental students. However, the sample consisted of 98 eligible dental students in clinical levels where the management of PWDs component is introduced, specifically fourth-, fifth-, and sixth-year students and interns. Dental interns are required to complete the internship as a prerequisite for graduation. The internship level is considered equivalent to a seventh-year dental student. These students have completed all the educational curricula, but their clinical experience is limited. At this stage, they are not yet licensed and must finish this year to receive their academic certification and become eligible to take the national exam required for obtaining a dental license. Preclinical dental students and graduates who had completed their internships were excluded.

### 2.2. Instruments

The questionnaire was designed after reviewing several survey instruments [15,16,17,18]. We developed the 41-item questionnaire, which comprised four sections:

The first section collected demographic information, asking about the students’ age, gender, and year of university study. The second section aimed to assess students’ experiences with PWDs and consisted of three items that measured their exposure levels to this population. The third section aimed to gather information about students’ education and training levels in managing PWDs through six items that inquired whether the students had theoretical knowledge of, had seen a clinical demonstration of, or had acquired community-based learning related to managing PWDs. Additionally, it asked about the effectiveness of their education and their desire for further education on the subject. 

The fourth section evaluated students’ attitudes toward PWDs using a modified version of the Dental Students’ Attitudes to the Handicapped Scale (DSATHS) [15]. This modification advocated for “person-first” language, replacing terms such as “handicapped patients” and “disfigured patients” with “PWDs” and “persons with facial disfigurement”. It contained two factors: factor 1 evaluated students’ views on their educational experiences and perceptions of their instructors (19 questions), while factor 2 measured students’ attitudes toward interpersonal and future interactions with PWDs (13 questions). Responses were rated on a 5-point Likert scale ranging from 1 (strongly agree) to 5 (strongly disagree). Although the questionnaire was based on previously validated survey instruments, face validity was determined by distributing the adapted questionnaire to five randomly selected dental students to ensure that the adapted questionnaire was clear, understandable, and relevant.

### 2.3. Data Collection Procedure

Eligible dental students and interns at King Faisal University were identified through the Admission and Registration Office at the Dental College. The participants’ email addresses were obtained from this office. All enrolled students were then approached via email. Each participant received an information sheet outlining the study’s purpose and an invitation letter. Participation was voluntary, anonymous, and entirely confidential. Online consent was obtained from each participant before they started the survey. The online questionnaire was distributed to 98 eligible dental students.

### 2.4. Data Analysis

Data were analyzed with SPSS version 24 (IBM^®^ SPSS^®^ Statistics, Armonk, NY, USA). Descriptive statistics were computed for numeric and categorical variables. Means and standard deviations (SD) were reported for continuous variables, and frequencies and percentages were reported for categorical variables. The reliability of the questionnaire was determined by using a reliability Cronbach’s alpha test for participants’ attitudes toward PWDs (Factor1; α = 0.75, Factor2; α = 0.81). Pearson’s chi-square test was applied to identify the relationship between participants’ university year levels and their levels of education and experience of managing PWDs. Fisher’s exact test was applied where the assumption of the Pearson chi-square test did not apply. 

Participants also completed the Dental Student Attitude toward the Handicapped Scale (DSATHS) questionnaire comprising 32 questions on a 5-point Likert scale (Strongly agree = 5, Agree = 4, Undecided = 3, Disagree = 2, Strongly disagree = 1). The assessment was based on questions from factor 1 (concerning educational experience and perceptions of instructors) and factor 2 (covering interpersonal and future interactions with PWDs). The scoring of unfavorable questions was reversed, and each participant’s final score was the sum of their scores for each item [15]. In the absence of a cut-off value for this scale, the percentile was used for categorization: based on the 33rd and 66th percentiles, total scores for “Attitude” (factors 1 and 2) were categorized into high, intermediate, and low (Factor 1: Minimum score = 43; Maximum score = 87) and (Factor 2: Minimum score = 21; Maximum score = 61) [19,20].

## 3. Results

Eighty-seven (87) of the 98 invited participants at the College of Dentistry, King Faisal University in Saudi Arabia, which only has male students, completed the survey, yielding a response rate of approximately 88.78%. The mean age of the participants was 23.38 years (±1.2 SD). Most of the respondents were undergraduate students in various years of study. A total of 23 students from the fourth year and 22 each from the fifth and sixth years responded. Students at the intern level (*n* = 20) made up nearly one-fourth of the total. Most (57.5%) of the participants reported that they were aware of PWDs in their environment (other than dental patients). Of the PWDs they knew, 29% were distant relatives, 19% were immediate family members, 15% were friends, and 10% were in the neighborhood. 

A statistically significant association between students’ university year levels and exposure to, or experience with, PWDs was evident (*p* value < 0.05). Sixth-year dental students, in particular, who made up 33.3% of the student total, reported that they were highly aware of and actively participated in outreach activities for PWDs. However, the 23% of students who merely knew about PWDs and did not know any, had not treated any, and were not taking part in any community-based activities for this population was comparable (Table 1).

A statistically significant correlation between students’ education or training levels in the treatment of PWDs and their university year levels was shown (*p* < 0.05). Overall, 27.6% of students had seen a theoretical and practical demonstration and undertaken community-based learning; 19.5% had undertaken a theoretical course only; 18.4% had had theoretical plus community-based learning; and 13.8% had theoretical plus practical course experience. Comparing students at different university year levels, those in the sixth year appear to be more prepared for and engaged in theoretical, practical, and community-based learning activities than those in other years (Table 2).

Most students (88.5%) want to gain additional information on how to manage PWDs, and 42.5% of the students reported that their dental education programs had not adequately prepared them to do so. The results showed that a statistically significant number of students wished to pursue further types of education (*p* < 0.05). Students at all levels showed more interest in clinical education to treat PWDs (70.1%) than in theoretical courses and outreach programs (29.9% and 34.5%, respectively) (Table 3).

A statistically significant correlation between students’ exposure to PWDs and their interpersonal or prospective interactions with them was found (*p* < 0.05) (Table 4). In comparison to students (2.3%) who had no previous exposure to PWDs, students (31.0%) who had some experience with PWDs generally had a more positive attitude toward interpersonal relationships and future interactions with this group (Table 4).

The results showed a statistically significant association between students’ education/training levels and their attitude to educational experiences and perceptions of instructors. The students’ attitude to their educational experiences and perceptions of instructors was high (16.1%) to intermediate (8.0%) when they participated in theoretical and practical demonstrations and community-based learning. Overall, students who had participated in any education or training addressing PWDs displayed a more positive attitude (26.4%) than those who had not (4.6%) (Table 5).

The findings showed a statistically significant correlation between students’ education/training and their future attitudes toward interpersonal contact with PWDs (*p* value < 0.05). The number of students with a positive attitude toward future interactions with PWDs was high (28.7%), and it was higher among those who had received any educational training in how to deal with PWDs than among those who had not (4.6%). However, the low attitude group had almost the same response. When given any educational training for treating a PWD, 29.9% of students expressed less interest in future interactions with PWDs (Table 6). 

The students’ university year level showed no association with their interpersonal and future encounters with PWDs (*p* value = 0.176). Similar percentages of students from all years were recorded, (33.3%), (34.5%), and (32.2%), respectively, in all three groups related to their attitude toward handling PWDs in the future.

## 4. Discussion

### 4.1. Exposure to PWDs 

The findings of this study indicate a statistically significant correlation between students’ exposure and their attitudes toward PWDs, including greater empathy, understanding, and willingness to provide care and support. The results were consistent with previous research suggesting that exposure to PWDs can positively influence individuals’ understanding of and attitudes toward individuals with disabilities [16,17,18,21]. In addition, a systematic review by Wang et al. found that contact with PWDs could lead to more positive attitudes toward them, which may be attributed to the likelihood that it “could help to reduce fear and anxiety and create a more balanced and realistic perspective about people with disabilities” [22]. However, a study conducted at Universidad Privada San Juan Bautista in Lima and Ica, Peru, which involved 100 dental interns and 75 dental professors, found that living with or treating patients with disabilities did not significantly improve their attitudes and perceptions regarding managing persons with disabilities (PWDs). The study recommended educational interventions incorporating real-life situations and specialized training to develop the necessary skills and knowledge for managing PWDs. These results suggested that exposure alone, without adequate training and support, was insufficient to effectively change attitudes toward the dental management of PWDs [23].

### 4.2. Education and Training for Managing People with Disabilities

Effective training programs can increase dental care professionals’ informed empathy. This study found a correlation between the respondents’ year levels and their education or training in the treatment of PWDs. Those in their sixth year had higher levels of theoretical engagement, practical demonstrations, and community-based learning activities. The findings also showed that students’ involvement in education or training to manage PWDs significantly influenced their attitudes positively toward their educational experiences and instructors. A statistically significant correlation was observed between students’ education or training and their future attitudes toward interpersonal interactions with PWDs. These findings confirm previous studies showing that educational interventions for dental students that aim to enhance the provision of care to PWDs increased their awareness of the special needs of PWDs, with the students indicating they would apply the knowledge in future practice [24,25,26]. In contrast, Holzinger et al. found that while dental students’ emotional reactions improved significantly after training, their social acceptance and willingness to treat PWDs did not. They suggested this might be due to students’ realization of their insufficient preparation, as their program consisted of five seminars and one practical course. To address this, they planned to expand the curriculum with more seminars and intensive extramural experiences to better impact students’ attitudes and readiness [27]. Similarly, Mac Giolla Phadraig et al. evaluated a comprehensive, blended learning Special Care Dentistry undergraduate program and found no statistically significant difference in student attitudes towards people with disabilities before and after the educational intervention. They noted that the ATDP scale, while robust, might not fully capture the nuances of attitude changes in this context and decided to consider using more specific scales designed for dental professionals in future evaluations [28]. Previous studies have examined the relationship between year level and attitude, and they have had conflicting results. For example, Rohani et al. found no significant difference in attitudes toward PWDs among students with different year levels [26], which was in contrast to the findings of Lee et al. [17].

### 4.3. Preparedness to Address PWDs’ Issues Effectively

Studies conducted in the United States and Malaysia have shown that students are not offered adequate training on treating PWDs, with dental schools dedicating different amounts of time to the subject [29,30]. A study in Ireland showed that only 27% of students believed their training was sufficient in providing treatment under supervision to patients with special needs, and only 19% felt confident in practicing special care dentistry (SCD) upon graduation [31]. A study conducted at two different universities—one in Scotland and one in Spain—found that 53.3% of final-year dental students felt that they lacked enough knowledge to properly treat all SCD patients upon graduation [32]. A study in Trinidad and Tobago revealed that only 41% of students felt confident in treating special needs patients upon graduation [33].

In alignment with previous studies, this study found that 42.5% of the participants believed that their dental education had not adequately prepared them to effectively address PWDs’ issues. Correspondingly, 88.5% expressed a desire for further education to better provide care for PWDs. This relatively high percentage indicates a need for comprehensive courses designed to improve students’ ability to address the needs of PWDs and enhance their competence. This would also be reflected in their evaluations of the training programs and trainers.

### 4.4. Bridging the Gap between Exposure, Training, and Preparedness

In Saudi Arabia, dental students and dental care providers have shown concern about their level of training, which may negatively affect their preparedness and willingness to help PWDs [16,34].

Similar to previous findings, 20.7% of the participants in the current study indicated that they had not received any training in managing PWDs. Additionally, 19.5% of all students reported that their education only provided them with theoretical knowledge of the issue. They had not received any clinical demonstrations or community-based education on PWDs. Students at all levels showed greater interest in receiving clinical education on treating PWDs (70.1%) than in theoretical courses and outreach programs (29.9% and 34.5%, respectively). This finding aligns with previous research indicating that students favor learning in clinical settings over didactic instruction [35,36].

Theoretical knowledge, while foundational, must be complemented with experiential expertise in managing PWDs. It must be achieved through direct patient interactions, hands-on training, and real-life simulations to foster a deeper understanding of patients’ perspectives, enhance empathy, improve care management, and transform students’ attitudes toward a patient-centered approach [37]. Increased clinical exposure to PWDs improves confidence, comfort, and willingness to treat this patient group [35,36]. Offering students real-world, hands-on experiences with PWDs through clinical rotations or community-based learning activities (which could be achieved outside the school environment) can significantly enhance their ability to provide holistic care [26,36,38].

An effective measurement tool regarding dental professionals’ attitudes toward PWDs would also provide numerous benefits. The tool could help identify students whose attitudes remain negative, thereby enabling teachers to intensify training efforts to improve their performance [15], and help standardize education programs for training students to address the dental care needs of PWDs. Currently, dental schools fail to provide such training due to full schedules covering other areas of dentistry, limited resources, and the perspective that the treatment of PWDs goes beyond the training of predoctoral dental students.

Notably, a more accurate assessment of dental students’ knowledge, skills, and attitudes toward treating PWDs could substantially improve training efforts. Research shows that improving training efforts regarding the treatment of PWDs “would make a difference in how future providers practice their profession, set up their practices, train their staff, and feel about treating special needs patients” [39]. Through the development of effective tools for assessing dental students’ attitudes toward PWDs, training can be optimized to ensure that they develop positive attitudes and the necessary skills to provide satisfactory services to this patient group.

Training institutions should also develop curricula based on evidence. Gathering such evidence requires the development of effective attitude assessment tools. One such evidence-based approach is the use of self-reflection components after exposure to training scenarios involving PWDs [40], which can help students develop positive attitudes toward this patient group. Students’ preparedness may vary depending on their understanding of disability issues, prior experience with disabilities, and the educational approach. This study’s results emphasized the need for structured teaching and increased exposure to the disabled community. Social interactions with and clinical exposure to PWDs enhanced students’ self-efficacy [21]. For instance, Krause et al. indicated that the more education dentists had received about providing care for patients with special needs, the better their attitudes were and the more likely they were to actually provide services for these patients [41]. Consequently, dental training institutions need to provide positive experiences involving opportunities for treating PWDs to improve students’ attitudes toward these patients [42].

### 4.5. Implications for Improving Dental Education and Care for PWDs in the Study Region

The findings of this study underscore the critical need to enhance dental education programs at King Faisal University to better prepare students for managing persons with disabilities (PWDs). Integrating comprehensive theoretical courses, practical demonstrations, and community-based learning experiences into the curriculum, while aligning with international accreditation standards [11,12], can significantly improve students’ skills and attitudes. This can be achieved by establishing effective partnerships with local organizations that support PWDs, including social institutions, general education schools, and colleges within the same university. These partnerships would provide valuable experience to dental students through interactions with PWDs and enhance their access to dental clinics.

This approach aligns with the new Saudi law issued on 22 August 2023, which regulates the rights of PWDs. The law emphasizes raising awareness about the rights of PWDs (Article 13) and their right to access health services (Article 9) [4]. Additionally, recruiting experienced and specialized faculty in dental management for PWDs and conducting continuous training programs for current faculty, which could be facilitated by introducing remote training as a quick and cost-effective approach [43], are essential steps. Establishing support systems within dental schools through partnerships with special education teachers and university faculty can help students manage the emotional challenges of treating PWDs.

Moreover, introducing simulation and technology, such as disability-simulating learning units and virtual patient modules, can enhance dental education for managing PWDs and improve dental students’ attitudes. This approach allows them to understand and manage various disabilities and dental conditions in a controlled environment, enhancing their competency without causing discomfort to real patients [44,45]. Improving the clinical environment for PWDs by integrating assistive tools, such as dimmed lighting, soothing sounds, and tactile stimuli, can further enhance communication, reduce anxiety, and mitigate negative behaviors, thus improving their overall dental care experience [46,47], which could enhance students’ experience dealing with PWDs.

Integrating teledentistry into dental school programs can equip future dentists with the skills needed to provide effective care for patients with disabilities, especially those who lack access to dental healthcare [48]. Furthermore, including the dental management for PWDs as a component of the national licensing exam for dental students would ensure that all graduating dentists have a standardized level of competency in this critical area, highlighting its importance within the dental profession.

Taking these factors into account can significantly improve dental education and care for PWDs in the study region and beyond.

### 4.6. Limitations and Future Directions

A limitation of this study is that the data were obtained exclusively from male participants at a single institution, which restricts the generalizability of the results to the broader population of Saudi dental students. Previous studies have shown varied results regarding gender differences in attitudes toward treating patients with special needs. For instance, some studies have found no notable differences between genders. A study conducted on dentists in Nigeria revealed no significant disparity across genders in their willingness to treat children with special needs [49]. Similarly, Tervo and Palmer reported no attitudinal differences by gender among health professional students in the United States [50]. On the other hand, a systematic review that assessed healthcare students’ and professionals’ attitudes found that female participants generally had more positive attitudes toward patients with physical disabilities than their male colleagues [51]. Conversely, a study conducted among dental care providers in different regions of Saudi Arabia indicated that male participants were more willing to treat patients with special needs compared to female participants [32]. Similarly, a study in the United States found that women dental students felt significantly less comfortable than men when treating these patients, which suggests that students might downplay their responsibility toward populations they feel less confident in treating [52]. These mixed findings highlight the need to include both genders in future research to gain insights into gender dynamics in attitudes toward PWDs. Despite these limitations, the broader context underscores the importance of this research. It is important to note that King Faisal University, which enrolls only male students, is the sole dental institution in the Al-Ahsa province providing comprehensive education in dentistry, including dental management for PWDs. Furthermore, according to the latest data from Saudi Arabia’s Ministry of Education, there are 4344 students with disabilities in general education in the Al-Ahsa governorate, highlighting the significant presence of this population in the study region [53]. The number is expected to be higher, as this statistic excludes those in specialized institutions such as rehabilitation centers or those not enrolled in schools.

## 5. Conclusions

In conclusion, this study emphasized the significance of exposure, education, and experiential learning in shaping dental students’ attitudes toward PWDs. Dental schools should prioritize structured teaching, increased exposure to the disabled community, and practical experiences to foster a patient-centered approach for future dental professionals. By doing so, we can create a more inclusive and compassionate dental care environment for PWDs.

## Figures and Tables

**Table 1 healthcare-12-01386-t001:** Student exposure and experience with PWDs across university year levels.

	BDS Year Level					
Students Who:	4th	5th	6th	Intern	Total	*p*-Value
Know and have treated a PWD	1 (1.1)	4 (4.6)	2 (2.3)	7 (8.0)	14 (16.1)	∞ 0.001 *
Know about and have engaged in any community outreach programs for PWDs	9 (10.3)	2 (2.3)	15 (17.2)	3 (3.4)	29 (33.3)	
Know PWDs, have treated them, and have engaged in any community outreach programs for PWDs	0 (0)	2 (2.3)	1 (1.1)	1 (1.1)	4 (4.6)	
Only know about PWDs	6 (6.9)	6 (6.9)	3 (3.4)	5 (5.7)	20 (23.0)	
None	7 (8.0)	8 (9.2)	1 (1.1)	4 (4.6)	20 (23.0)	
Total	23 (26.4)	22 (25.3)	22 (25.3)	20 (23.0)		

∞ Fisher’s exact test applied; PWDs: persons with disabilities; * statistical significance (*p* < 0.05).

**Table 2 healthcare-12-01386-t002:** Students’ education/training in the treatment of PWDs across university year levels.

	BDS Year Level *n* (%)				Total	*p* Value
Students Who:	4th	5th	6th	Intern		
Had only taken a theoretical course	3 (3.4)	6 (6.9)	1 (1.1)	7 (8.0)	17 (19.5)	∞ 0.001 *
Had taken a theoretical course and seen a practical demonstration	2 (2.3)	5 (5.7)	3 (3.4)	2 (2.3)	12 (13.8)	
Had theoretical and community-based learning	3 (3.4)	2 (2.3)	7 (8.0)	4 (4.6)	16 (18.4)	
Had seen a theoretical and practical demonstration and undertaken community-based learning.	9 (10.3)	3 (3.4)	11 (12.6)	1 (1.1)	24 (27.6)	
None	6 (6.9)	6 (6.9)	0 (0)	6 (6.9)	18 (20.7)	
Total	23 (26.4)	22 (25.3)	22 (25.3)	20 (23.0)		

∞ Fisher’s exact test applied; * statistical significance (*p* < 0.05).

**Table 3 healthcare-12-01386-t003:** Comparison of students’ education/training/additional requirements for treating PWDs and university year levels.

Year Level							
Variables	Choices	4th Year BDS *n* (%)	5th Year BDS *n* (%)	6th Year BDS *n* (%)	Internship BDS *n* (%)	Total *n* (%)	*p* Value
Students who reported that their dental education had prepared them effectively to treat PWD	Yes	9 (10.3)	7 (8.0)	13 (14.9)	4 (4.6)	33 (37.9)	∞ 0.238
	No	10 (11.5)	11 (12.6)	5 (5.7)	11 (12.6)	37 (42.5)	
	Don’t know	4 (4.6)	4 (4.6)	4 (4.6)	5 (5.7)	17 (19.5)	
Would students like more education concerning the treatment of PWD?	Yes	23 (26.4)	15 (17.2)	21 (24.0)	18 (20.7)	77 (88.5)	∞ 0.004 *
	No	0 (0)	7 (8.0)	1 (1.1)	2 (2.3)	10 (11.5)	
Which type of education do students want to engage in?							∞
Didactic (courses, lectures)	Yes	5 (5.7)	6 (6.9)	9 (10.3)	6 (6.9)	26 (29.9)	0.557 ^≅^
	No	18 (20.7)	16 (18.4)	13 (14.9)	14 (16.1)	61 (70.1)	
Clinical	Yes	22 (25.3)	10 (11.5)	16 (18.4)	13 (14.9)	61 (70.1)	^≅^ 0.003 *
	No	1 (1.1)	12 (13.8)	6 (6.9)	7 (8.0)	26 (29.9)	
Community-based dental education	Yes	5 (5.7)	6 (6.9)	11 (12.6)	8 (9.2)	30 (34.5)	^≅^ 0.189
	No	18 (20.7)	16 (18.4)	11 (12.6)	12 (13.8)	57 (65.5)	

∞ Fisher’s exact test applied; ^≅^ Pearson’s chi-square test applied; * statistical significance (*p* < 0.05); PWDs: persons with disabilities.

**Table 4 healthcare-12-01386-t004:** Comparison of students’ levels of exposure to PWDs with their attitudes to interpersonal and future interactions with this group.

Students with Different Levels of Exposure to PWDs							Total *n* (%)	*p* Value
Variable	Categories	Students Who Know and Have Treated a PWD *n* (%)	Students Who Know a PWD and Have Engaged in Any Volunteering for PWDs *n* (%)	Students, Who Know PWDs, Have Treated Them, and Have Engaged in Volunteering Activities for PWDs *n* (%)	Students Who Know a PWD Only *n* (%)	None *n* (%)		
Interpersonal and future interactions with PWD.	High attitude	10 (11.5)	7 (8.0)	2 (2.3)	8 (9.2)	2 (2.3)	29 (33.3)	∞ <0.001 *
	Intermediate attitude	3 (3.4)	7 (8.0)	0 (0.0)	10 (11.5)	10 (11.5)	30 (34.5)	
	Low attitude	1 (1.1)	15 (17.2)	2 (2.3)	2 (2.3)	8 (9.2)	28 (32.2)	
	Total	14 (16.1)	29 (33.3)	4 (4.6)	20 (23.0)	20 (23.0)		

∞ Fisher’s exact test applied; PWDs: persons with disabilities; * statistical significance (*p* < 0.05).

**Table 5 healthcare-12-01386-t005:** Comparison of students’ education/training with their attitude to educational experiences and perceptions of instructors.

		Students’ Education and Training Levels for Managing PWDs					Total *n* (%)	*p* Value
Variable	Categories	Students Who Had Only Taken a Theoretical Course *n* (%)	Students Who Had Taken a Theoretical Course and Seen a Practical Demonstration *n* (%)	Students Who Had Had Theoretical and Community-Based Learning *n* (%)	Students Who Had Seen a Theoretical and Practical Demonstration and Engaged in Community-Based Learning *n* (%)	None *n* (%)		
Students’ responses to attitude statements regarding their educational experiences and perceptions of instructors	High attitude	0 (0.0)	3 (3.4)	6 (6.9)	14 (16.1)	4 (4.6)	27 (31.0)	∞ 0.003 *
	Intermediate attitude	6 (6.9)	5 (5.7)	5 (5.7)	7 (8.0)	9 (10.3)	32 (36.8)	
	Low attitude	11 (12.6)	4 (4.6)	5 (5.7)	3 (3.4)	5 (5.7)	28 (32.2)	
	Total	17 (19.5)	12 (13.8)	16 (18.4)	24 (27.6)	18 (20.7)		

∞ Fisher’s exact test applied; * statistical significance (*p* < 0.05); PWDs: persons with disabilities.

**Table 6 healthcare-12-01386-t006:** Comparison of students’ education/training with their attitudes to interpersonal and future interactions with PWDs.

Students’ Education/Training Levels for Managing PWDs							Total *n* (%)	*p* Value
Variable	Categories	Had Only Theoretical Knowledge *n* (%)	Theoretical Course and Practical Demonstration *n* (%)	Theoretical and Community-Based Learning *n* (%)	Theoretical and Practical Demonstration and Community-Based Learning *n* (%)	None *n* (%)		
Interpersonal and future interactions with PWDs	High attitude	7 (8.0)	6 (6.9)	6 (6.9)	6 (6.9)	4 (4.6)	29 (33.3)	∞ 0.024 *
	Intermediate attitude	7 (8.0)	1 (1.1)	4 (4.6)	6 (6.9)	12 (13.8)	30 (34.5)	
	Low attitude	3 (8.0)	5 (5.7)	6 (6.9)	12 (13.8)	2 (2.3)	28 (32.2)	
	Total	17 (8.0)	12 (13.8)	16 (18.4)	24 (27.6)	18 (20.7)		

∞ Fisher’s exact test applied; * statistical significance (*p* < 0.05); PWDs: persons with disabilities.

## Data Availability

The data that support the findings of this study are available from the corresponding author upon reasonable request.

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
