# Peer review of "Bridging the Gap: Dental Students’ Attitudes toward Persons with Disabilities (PWDs)"

_healthcare, 2024, doi:10.3390/healthcare12141386_

Round 1
Reviewer 1 Report
Comments and Suggestions for Authors
Dear authors
It is an interesting findings providing insight on the structured education curricula.
I have gone through the draft and satisfied with its quality. It was well written and presented.
The authors have ensured the flow of thoughts through the draft. The points presented in the draft was cohesive in between paragraphs.
Methodology section was adequately prepared. The questionnaires were validated before application in their study. The selection of sample was carefully done.
Results were presented in concise manner. All tables were interpreted. Presentation of statistical data was adequate. This followed by discussion where each pertinent points in the result section had been adequately discussed.
Good conclusion where each specific objective has been explained.
Referencing style was consistent and standardized.
Only minor issues on the sample that require some explanation that 98 students were the enrollment of Dental Faculty at King Faisal Univ.

Author Response
Thank you for your thorough review and positive feedback on our manuscript. We appreciate your recognition of the quality and structure of our work, as well as your comments on the methodology, results, discussion, and referencing.
Regarding the minor issue you highlighted about the sample size, we would like to provide further clarification as addressed in the revised manuscript. The initial pool included a total of 166 dental students. However, the sample consisted of 98 eligible dental students in clinical levels where the management of PWDs component is introduced, specifically fourth, fifth, and sixth-year, and internship students. Preclinical dental students and graduates who had completed their internships were excluded.
Reviewer 2 Report
Comments and Suggestions for Authors
Unfortunately, the manuscript entitled "Bridging the Gap: Dental Students’ Attitudes toward Persons with Disabilities (PWDs)" by Asiri et al. lacks enough clinical significance and does not meet the qualifications necessary for publication in this journal. A number of major limitations include enrollment of only male subjects, single-center study, convenience sampling method, and self-report measures. However, I have provided a few significant comments to revise and improve the manuscript before sending it to another journal:
1. The authors should provide a more in-depth discussion of the existing literature, including potential conflicting results and limitations of prior studies, to contextualize the current findings within the broader scientific landscape.
2. The study found a strong correlation between students' exposure to PWDs and their attitudes, as well as the impact of education and training. It is crucial to discuss the strength of these correlations and any potential confounding variables that could influence the results, as this will ensure the study is thorough and comprehensive.
3. The study provides valuable insights into the attitudes and preparedness of dental students in Saudi Arabia. For a more impactful study, it is important to discuss the implications of the findings on improving dental education and care for people with disabilities in the region.
Author Response
Reviewer 2
General Comment:
‘’Unfortunately, the manuscript entitled "Bridging the Gap: Dental Students’ Attitudes toward Persons with Disabilities (PWDs)" by Asiri et al. lacks enough clinical significance and does not meet the qualifications necessary for publication in this journal. A number of major limitations include enrollment of only male subjects, single-center study, convenience sampling method, and self-report measures. However, I have provided a few significant comments to revise and improve the manuscript before sending it to another journal:’’
Response:
We appreciate the opportunity to clarify the details regarding our study limitations. While acknowledging the limitations noted by the reviewer, such as the enrollment of only male subjects and the single-center study design, we further emphasized that King Faisal University is the sole dental institution in the Al-Ahsa province, providing comprehensive education in dentistry, including special care dentistry. We also highlighted the significant presence of students with disabilities in the region, as indicated by data from the Ministry of Education. Based on the latest data from Saudi Arabia’s Ministry of Education (2023), there are 4,344 students with disabilities enrolled in general education in the Al-Ahsa governorate, highlighting the significant presence of this population in the study region. The number is expected to be higher, as this statistic excludes those in specialized institutions such as rehabilitation centers or those not enrolled in schools.
Additionally, we point out the alignment with the new Saudi law issued on August 22, 2023, which regulates the rights of PWDs. The law emphasizes raising awareness about the rights of PWDs (Article 13) and their right to access health services (Article 9) [4]. This new regulation aligns with the objectives of our study and stresses the importance of improving dental education, training, and attitudes to better serve PWDs in the region. We believe our research could provide valuable insights and draw the attention of healthcare policymakers to improving healthcare delivery for PWDs. By highlighting gaps in current educational practices and proposing evidence-based interventions focusing on dental students' attitudes and dental management for PWDs, our study aims for systemic changes that can lead to better healthcare outcomes for PWDs. This aligns with the journal's special issue thematic focus on healthcare policy, inequity, and systems research.
Furthermore, we included in our limitations a further discussion regarding previous studies that have shown varied results regarding gender differences in attitudes toward treating patients with special needs. For instance, some studies have found no notable differences between genders. A study conducted on dentists in Nigeria revealed no significant disparity across genders in their willingness to treat children with special needs [48]. Similarly, Tervo and Palmer reported no attitudinal differences by gender among health professional students in the United States [49]. On the other hand, a systematic review in 2012 that assessed healthcare students' and professionals' attitudes found that female participants generally had more positive attitudes toward patients with physical disabilities than their male colleagues [50]. Conversely, a study conducted among dental care providers in different regions of Saudi Arabia indicated that male participants were more willing to treat patients with special needs compared to female participants [31]. Similarly, a study in the United States found that women dental students felt significantly less comfortable than men when treating these patients, which suggests that students might downplay their responsibility toward populations they feel less confident in treating [51].
Significant Comments to Revise and Improve the Manuscript:
Comment 1:
"The authors should provide a more in-depth discussion of the existing literature, including potential conflicting results and limitations of prior studies, to contextualize the current findings within the broader scientific landscape."
Response:
Thank you for your constructive comment. We have included further studies and discussions to provide a more in-depth analysis of the existing literature, addressing potential conflicting results and limitations of prior studies. These additions are highlighted in the revised manuscript using track changes.
Comment 2:
- "The study found a strong correlation between students' exposure to PWDs and their attitudes, as well as the impact of education and training. It is crucial to discuss the strength of these correlations and any potential confounding variables that could influence the results, as this will ensure the study is thorough and comprehensive."
Response:
Thank you for your valuable comment. The term "strong correlation" was an oversight. Our analysis indicates statistically significant associations rather than specifying the strength of these correlations. Thank you for highlighting this issue. We have revised the text to state "statistically significant association" instead of "strong correlation." This change ensures that our terminology is precise and accurately reflects the statistical data presented.
Comment 3:
- "The study provides valuable insights into the attitudes and preparedness of dental students in Saudi Arabia. For a more impactful study, it is important to discuss the implications of the findings on improving dental education and care for people with disabilities in the region."
Response:
Thank you for your insightful comment. We have expanded the discussion to address the implications of our findings on improving dental education and care for people with disabilities in the region. These revisions are highlighted in the revised manuscript using track changes.
Reviewer 3 Report
Comments and Suggestions for Authors
Author Response
Comment 1:
Page 2: "In Saudi Arabia... there should not be a period after the brackets, but the sentence should continue."
Response:
Thank you for pointing this out. We have corrected the sentence on Page 2 to remove the period after the brackets so that the sentence continues appropriately in the revised manuscript.
Comment 2:
Please note in the "Methods" section, when mentioning the Ethical Approval, state in which city and country King Faisal University is located.
Response:
We appreciate your suggestion. In the "Methods" section, we have updated the manuscript to specify that King Faisal University is located in Al-Ahsa, Saudi Arabia when mentioning the Ethical Approval.
Comment 3:
In the "Methods" section ... "We designed a cross-sectional" is missing the word "study."
Response:
Thank you for noticing this. We have corrected the sentence in the "Methods" section to "We designed a cross-sectional study" in the revised manuscript.
Comment 4:
- ‘’Under the study's limitations, the authors stated that the respondents were exclusively male. I think that this is the biggest shortcoming of this study. Women would undoubtedly give different answers to the questions, and I suggest you either get involved in this research or do new research that would include women’’.
Response:
Thank you for your valuable feedback and for highlighting the importance of including gender perspectives in our study. We appreciate your suggestion to involve female participants in future research. We have addressed this concern by including in our limitations a further discussion regarding previous studies that have shown varied results regarding gender differences in attitudes toward treating patients with special needs. This addition acknowledges the gender-related limitations of our study and emphasizes the importance of including both male and female perspectives in future research to better understand gender dynamics in attitudes toward PWDs.
Comment 5:
- Please explain what "outreach program "in Table 1 means.
Response: Thank you for your comment. In the revised version of our manuscript, we have clarified that "outreach program" refers to "community outreach programs for PWDs." These programs involve dental students engaging in activities that extend beyond their formal education, including volunteer activities aimed at providing services and/or education to persons with disabilities within the community.
Comment 6:
- In Table 3, considering the variable "Students who reported that their dental education had
prepared them effectively to treat PDW, "there are considerable discrepancies between 6th-year BDS and internship. Can you explain the reason for that? Did the university change some programs?
Response:
Thank you for your insightful comment. In the KFU dental program, the internship level is considered equivalent to a seventh-year dental student. These students have successfully completed all the educational curricula, but their clinical experience is limited. At this stage, they are not yet licensed and are required to finish this year to receive their academic certification and apply for the licensure exam. In the revised version of our manuscript, we have clarified this ; see the track changes in the Study Participants section.The discrepancies observed might be attributed to the increased exposure and autonomy during the internship year, where students face real-life clinical situations with minimal supervision. This could lead to a more critical self-assessment and a lower perception of preparedness compared to sixth-year students, who might still be under more structured and supportive clinical environments. However, as shown in Table 3, the year level was not statistically significant.
Comment 7:
- In Table 5, the main title is "Students' Education and Training Levels for Treating PWDs," and Table 6. "Students' Education/Training Levels for Managing PWDs. " Can you explain the difference
between treating and managing PWDs?
Response:
Thank you for your valuable comment .The terms "treating" and "managing" can often be used interchangeably; however, in this context, "managing" is a more comprehensive term. It encompasses not only the clinical treatment but also the theoretical, clinical, and community-based components necessary for managing PWDs. Therefore, using "managing" in these tables is appropriate as it provides a more holistic view of the education and training levels required for dental students to work effectively with PWDs. In the revised manuscript, we have changed the terminology for the table 5 to "managing" to better reflect the comprehensive approach used.
Comment 8:
- Apart from the fact that they are mentioned as a part of the respondents, nowhere else in the study is there a discussion about the interns and their knowledge, attitudes, and education... that part is sorely lacking
Response:
Thank you for your comment. As explained in the previous comment, in the King Faisal University (KFU) dental program, interns are regarded as equivalent to seventh-year dental students. These interns have finished all their theoretical coursework but have not yet gained extensive practical, hands-on experience. They are not yet licensed dentists. They must complete this internship year to receive their degree and become eligible to take the national exam required for obtaining a dental license. The revised manuscript has been updated to clearly explain the status and role of interns in the dental program. Interns are included in the group of students being studied.
Reviewer 4 Report
Comments and Suggestions for Authors
Dear Authors,
Thank You for a pleasure to read Your manuscript.
I have several comments and offers to improve Your article.
Firstly, the question about the determination of “Persons with disabilities” presents. It includes many conditions and diseases. Could You, please, make a point?
M&Ms
Section 2.1 Why did You use these groups of students for questionary?
Sincerely, Reviewer
Author Response
Comment 1:
Firstly, the question about the determination of “Persons with disabilities” presents. It includes many conditions and diseases. Could You, please, make a point?
Response:
Thank you for your question and comment. In the introduction, we have clarified the definition of "Persons with disabilities" in the revised manuscript. (Intro)
Comment 2:
2: Section 2.1 Why did you use these groups of students for the questionnaire?
Response:
Thank you for your question. The groups of students selected for the questionnaire—specifically fourth, fifth, sixth-year students, and interns—were chosen because the dental management of Persons with Disabilities (PWDs) component is introduced at these clinical levels. This is highlighted in the revised manuscript.
Round 2
Reviewer 2 Report
Comments and Suggestions for Authors
Unfortunately, the author's explanation was unsatisfactory, and this study does not meet the qualifications necessary for publication in this journal.